# The Distribution of Circulating Tumor Cells Is Different in Metastatic Lobular Compared to Ductal Carcinoma of the Breast—Long-Term Prognostic Significance

**DOI:** 10.3390/cells9071718

**Published:** 2020-07-17

**Authors:** Ulrik Narbe, Pär-Ola Bendahl, Kristina Aaltonen, Mårten Fernö, Carina Forsare, Charlotte Levin Tykjær Jørgensen, Anna-Maria Larsson, Lisa Rydén

**Affiliations:** 1Department of Clinical Sciences, Division of Oncology, Lund University, SE-223 81 Lund, Sweden; ulrik.narbe@med.lu.se (U.N.); par-ola.bendahl@med.lu.se (P.-O.B.); marten.ferno@med.lu.se (M.F.); carina.forsare@med.lu.se (C.F.); charlotte.levin_tykjaer_jorgensen@med.lu.se (C.L.T.J.); Anna-Maria.Larsson@med.lu.se (A.-M.L.); 2Department of Oncology, Växjö Central Hospital, SE-352 34 Växjö, Sweden; 3Department of Laboratory Medicine, Division of Translational Cancer Research, Lund University, SE-223 81 Lund, Sweden; kristina.aaltonen@med.lu.se; 4Department of Hematology, Oncology and Radiation Physics, Skåne University Hospital, SE-221 85 Lund, Sweden; 5Department of Clinical Sciences, Division of Surgery, Lund University, SE-223 81 Lund, Sweden; 6Department of Surgery, Skåne University Hospital, SE-221 85 Lund, Sweden

**Keywords:** circulating tumor cells, CTC cluster, invasive lobular carcinoma, invasive ductal carcinoma of no special type, cancer antigen 15-3

## Abstract

Background: Invasive lobular carcinoma (ILC) has distinguishing features when compared to invasive ductal carcinoma of no special type (NST). In this study, we explored the distributional and prognostic characteristics of circulating tumor cells (CTCs) in metastatic ILC and NST. Materials and methods: Patients were included in an observational trial (ClinicalTrials.gov NCT01322893) with ILC (*n* = 28) and NST (*n* = 111). CTC count (number/7.5 mL blood) was evaluated with serial sampling (CellSearch). The primary endpoint was progression-free survival (PFS). Results: The CTC counts were higher in ILC (median 70) than in NST cases (median 2) at baseline (*p* < 0.001). The evidence for ≥5 CTCs as a prognostic factor for PFS in ILC was weak, but stronger with higher cut-offs (CTC ≥ 20: hazard ratio (HR) 3.0, *p* = 0.01) (CTC ≥ 80: HR 3.6, *p* = 0.004). In NST, however, the prognostic effect of CTCs ≥5 was strong. Decline in CTC count from baseline to three months was associated with improved prognosis in ILC and NST. Conclusions: The number of CTCs is higher in ILC than in NST, implying that a higher CTC cut-off could be considered for ILC when applying the CellSearch technique.

## 1. Introduction

Breast cancer (BC) is the most common of all malignancies in women. According to data from the World Health Organization, 2.1 million new cases were diagnosed worldwide in 2018. Approximately 20%–30% of all BCs at some point present as metastatic breast cancer (MBC) with dissemination to distant organs. MBC is a chronic and often fatal disease, resulting in >500,000 BC-related deaths annually [1]. In recent decades, modern BC care has considerably progressed, but unmet needs still remain, especially for MBC. Research focusing on new prognostic and monitoring factors, as well as improving those used today, is important to optimize clinical strategies and treatments for those living with MBC.

Clinical examination (CE) and diagnostic imaging (DI) are the current standard methods for monitoring MBC. Treatment response evaluation with DI using the Response Evaluation Criteria In Solid Tumors (RECIST) criteria [2] in bone tissue is difficult and considered more unreliable than in other metastatic sites (e.g., liver, lungs, and lymph nodes) [3,4]. Therefore, new complementary monitoring methods, for example, liquid biopsies, are especially important in MBC with solitary bone metastases. Among the liquid biopsy techniques, the serum tumor marker cancer antigen 15-3 (CA 15-3) is a BC-associated tumor marker with putative monitoring potential and might also harbor prognostic information. However, its clinical usefulness and reliability have not been fully validated, and no clear cut-off value has been established [5,6].

Circulating tumor cells (CTCs) have been extensively studied and have repeatedly been shown to carry prognostic and monitoring information in MBC. A CTC count of ≥5 cells per 7.5 mL blood is a previously validated cut-off in MBC for the CellSearch technique [7,8,9,10]. Prognostic and monitoring data from preexisting MBC studies demonstrate that CTCs have a clinical validity superior to CA 15-3 [7,11]. 

CTC clusters are defined as a group of two or more tumor cells with strong cell–cell adhesion [12]. Studies suggested that the presence of CTC clusters is a negative prognostic factor in MBC, and could be of potential prognostic significance in addition to single CTCs [13,14]. Findings in a preclinical study based on mouse models also indicated that the metastatic capacity of CTC clusters might be up to 50-fold higher compared to single CTCs [12].

A strong correlation between CTC and DI for predicting progressive disease was found, and some studies suggested that CTCs can detect disease progression before DI and could thus be a valid monitoring tool [15,16]. 

The most frequent histopathological type of BC is invasive ductal carcinoma of no special type (NST) (~80%), and invasive lobular carcinoma (ILC) is the second most common type (~10%) [17]. ILC has, compared to NST, many distinguishing clinicopathological [18,19,20,21,22] and genomic [23,24,25] features. ILCs tend to have a higher incidence of late recurrence (>10 years past primary diagnosis), a somewhat different pattern of recurrence sites and metastatic dissemination, with a higher frequency of solitary bone metastases and atypical metastatic sites (e.g., orbitae, peritoneum, pituitary and adrenal glands, and gynecological and gastrointestinal organs) [20,26,27]. Despite all the differences, the clinical management of these two types of BC is similar, and current treatment guidelines are based on results from BC trials where NSTs compose a majority of cases. Studies exclusively focusing on metastatic ILC are to date very sparse.

The results from various studies investigating differences in CTC counts between BC subtypes are ambiguous. A high CTC count (≥5) in MBC was associated with triple-negative (TN) and luminal A-like subtype [7,14]. However, a meta-analysis including 24 studies showed that a high CTC count (≥5) was more common in HER2-positive than in TN and luminal A- and B-like MBC [28]. Two MBC studies revealed a higher prevalence of ≥1 CTC and a higher CTC count in the ILC than the NST subgroup, but no survival data presenting the prognostic importance of CTCs in ILC and NST cohorts were provided [29,30] and the prognostic value of a higher number of CTCs in ILC is thus hitherto unknown. Putative explanations to the higher number of CTCs in ILC include the predominance of a luminal A-like phenotype [29] and loss of cell-to-cell adhesion in this histopathological tumor type [30].

In this study, we aimed to explore the distribution and prognostic significance of CTC count and CTC clusters in patients with metastatic ILC vs. NST in a prospective monitoring trial exclusively including patients scheduled for first-line systemic therapy subject to longitudinal CTC sampling during six months with available long-term survival data. Secondary aims were to evaluate CTCs in relation to standard monitoring methods as well as CA 15-3 and the prognostic value of change of CTC level.

## 2. Materials and Methods

### 2.1. Patients and Study Design

The study was based on a previously reported prospective observational MBC cohort comprising mixed histopathological types (*n* = 156) [14,31], including a subpopulation of 28 patients with pure ILC and 111 patients with pure NST (Figure 1). Patients diagnosed with MBC and scheduled for first-line systemic treatment at Skåne University Hospital and Halmstad County Hospital, Sweden, between April 2011 and June 2016 were enrolled in a trial (ClinicalTrials.gov NCT01322893) conducted by the Department of Oncology and Pathology at Lund University, Sweden. Inclusion criteria: age older than eighteen years, Eastern Cooperative Oncology Group (ECOG) performance status score 0–2, predicted life expectancy longer than two months, and available data on histopathological subtype. Exclusion criteria: prior systemic therapy for metastatic breast cancer, inability to provide informed consent, and any diagnosis of malignancy within 5 years before inclusion [14]. Patient and tumor characteristics and follow-up (FU) data were retrieved from case report forms (CRFs) and clinical records. Follow-up data were updated as of 23 April 2019, after which the database was locked. The study was approved by the Lund University Ethics Committee (LU 2010/135).

### 2.2. Detection of CTCs and CTC Clusters 

Blood samples were collected in 10 mL CellSave Preservation tubes (Menarini Silicon Biosystems, Bologna, Italy), stored between 15 and 30 °C, and processed within 96 h of collection. CTCs were isolated and enumerated using the Food and Drug Administration-approved CellSearch system (Menarini Silicon Biosystems, Florence, Italy), as described in detail previously [10,32]. Two investigators certified in the CellSearch technology independently evaluated all images within the generated galleries for events. Any event for which the assessment differed between the investigators was re-evaluated until consensus was reached. CTC count and number of CTC clusters per 7.5 mL blood were evaluated at baseline (BL) and during treatment at 1, 3, and 6 months. 

A CTC count of ≥5 was considered high, and additional exploratory cut-offs of ≥20 and ≥80, based on previous studies by Botteri et al. [33] and Peeters et al. [29], were also evaluated.

CTC clusters were defined as groups consisting of ≥2 CTCs clustered together, with non-overlapping nuclei. A blood sample was considered positive for CTC clusters if ≥1 CTC cluster was detected. Two independent assessors evaluated CTC clusters in CTC galleries exported from the CellTracks Analyzer II system, as described previously [31]. Photomicrographs of CTC clusters are provided in Appendix A.

### 2.3. CA 15-3

The serum marker CA 15-3 was analyzed at the Department of Clinical Chemistry at Skåne University Hospital with an accredited method used in clinical practice (CA 15-3 on Cobas, NPU01449, Roche, Basel, Switzerland). CA 15-3 values ≥30 U/mL were considered high [34]. Additional experimental cut-offs of ≥100, ≥200, and ≥400 U/mL (predefined) were also analyzed.

### 2.4. Monitoring Procedure of CTCs and CA15-3 for Detection of Early Progression in MBC

The patients underwent structured evaluation via CE and DI (standard monitoring methods) at least every 3 months according to a prespecified study protocol during follow-up, which was continued after the serial blood sampling. Progression vs. non-progression was defined according to clinical practice based on CE and DI, using modified RECIST 1.1 criteria [2] (progression: progressive disease vs. non-progression: stable disease, partial response, or complete response). Aiming at detection of early progression during 0–12 months by CTCs and CA15-3 in relation to routinely diagnosed progression, CTC progression was defined as either (1) an increase in CTC count from <5 to ≥5 or, for those patients with ≥5 CTCs at BL, (2) an increase ≥25% (predefined) in the number of detected CTCs, between two adjacent time-points (BL and 1 month, 1 and 3 months, or 3 and 6 months). Similarly, an increase in CA 15-3 levels of ≥25% between two time-points was classified as CA 15-3 progression [34]. 

The change in level of CTCs and CA 15-3 during 0–6 months of FU were related to progression confirmed by standard monitoring methods, which were restricted to 0–12 months of FU. 

### 2.5. Statistics

Evidence for differences in categorical variables, including patient and tumor characteristics, and CTC and CA 15-3 levels between the histopathological types (ILC vs. NST) were evaluated using Pearson’s chi-squared test or, if expected counts under the null hypothesis were lower than 5 in one or more of the cells of a contingency table, Fisher’s exact test. Ordinal data were evaluated using Pearson’s chi-squared test for trends, also known as the linear-by-linear association test, whereas variables measured on a continuous scale were evaluated using the Mann–Whitney U test.

The primary endpoint was progression-free survival (PFS) and the secondary endpoint was overall survival (OS) in relation to CTC counts. For each patient, FU time was calculated from the date of the first blood draw to progression or death from any cause. Patients without disease progression or those who were still alive at the last FU date were censored for PFS and OS, respectively. Kaplan–Meier plots and the log-rank test were used to illustrate and compare survival between subgroups. Cox proportional hazards regression was used for the estimation of hazard ratios (HRs), and proportional hazards assumptions were checked graphically. Landmark analysis was used to study change in CTC status from baseline to 3 months in relation to PFS and OS from 3 months and onwards.

To explore the effect on outcome by dynamic changes in CTC status, Cox models with a time-dependent covariate were applied to estimate the effects of the longitudinally measured binary variable ≥5 CTCs on PFS and OS. The follow-up for patients whose CTC status changed during follow-up was split into 2–4 non-overlapping episodes with constant CTC status, for example, two episodes if the CTC status changed at a single time point and four if it changed at all of the three follow-up time points 1, 3, and 6 months. A single episode was sufficient for patients with the same CTC status at BL and at all follow-up sampling occasions. Missing CTC status at follow-up visits was imputed using the last observation carried forward principle. The HR for ≥5 CTCs in models of this kind should be interpreted as the ratio of the progression incidence during episodes with ≥5 CTCs to that of episodes with <5 CTCs.

The REMARK recommendations for reporting of tumor marker studies were followed [35]. IBM SPSS Statistics (version 25.0, IBM Inc., Armonk, NY, USA) and Stata version 16 (StataCorp, College Station, TX, USA) were used for statistical calculations.

### 2.6. PAM 50 Subtyping

RNA was isolated from archival formalin-fixed, paraffin-embedded tumor tissue using AllPrep DNA/RNA FFPE kit (Qiagen, Hilden, Germany) according to the instructions provided by the manufacturer. Samples were run on the NanoString Breast Cancer 360 (BC360) assay on the nCounter Sprint Profiler (NanoString Technologies, Seattle, WA, USA). Based upon the 50 gene expression signature detailed by Parker et al. [36], PAM50 breast cancer intrinsic subtyping analysis was completed at NanoString Technologies, Seattle, WA, USA, and samples were classified into luminal A, luminal B, HER2-enriched, and basal-like subtypes.

## 3. Results

### 3.1. Patient Outcome

The median FU time was 49 (27–93) months. At the time of data-base lock (April 2019), 9% of the patients were progression-free and 30% were alive (Table 1). The progression and mortality rates for patients with ILC compared to NST were similar over the whole FU period (PFS: HR 0.89, 95% confidence interval (CI) 0.57–1.4, *p* = 0.59; OS: HR 0.99, 95% CI 0.60–1.6, *p* = 0.96), but during the first two years, PFS and OS were higher among those with ILC (Appendix A). Compared to NST, metastatic ILC cases were more often luminal A subtype (71% vs. 31%; *p* = 0.001) and axillary node-positive (92% vs. 64%; *p* = 0.007) at the time of primary diagnosis. They presented with three or more metastatic sites in 14% vs. 32% (*p* = 0.10), visceral metastases in 29% vs. 65% (*p* = 0.001), and solitary bone metastases in 39% vs. 22% (*p* = 0.09) for metastatic ILC and NST cases, respectively. First-line systemic treatment was similar among both ILC and NST cases (endocrine: 46% vs. 39%, chemotherapy: 46% vs. 52%, and HER2-targeted: 7% vs. 8%; Table 1).

### 3.2. CTC, CTC Cluster, and CA 15-3 Analyses

#### 3.2.1. Descriptive Data

A CTC count of five or more was more common at BL in ILC than in NST cases (22/28 vs. 49/107; *p* = 0.003), displayed in Figure 2, a difference corresponding to an odds ratio (OR) for CTC positivity of 4.3 (95% CI: 1.6–12). The same pattern for an increased risk of CTC positivity in ILC was also observed after adjustment for the PAM50 subtype (luminal A vs. non-luminal A) of the primary tumor (OR ILC vs. NST: 5.9; 95% CI: 1.8–20; *p* = 0.004). The evidence was strong (*p* < 0.001) for a difference in the distribution of CTC counts between ILC (median 70, interquartile range (IQR) 121) and NST cases (median 2, IQR 32) at BL, and the presence of CTC clusters was also more common among ILC cases (36% vs. 18%, *p* = 0.07; Table 2). CTC count and the number of CTC clusters declined in both ILC (median count 4, IQR 14; clusters: 4%) and NST cases (median count 0, IQR 7; clusters: 12%) after one month of systemic treatment. There was strong evidence of a decline in CTC count from BL to all time points for ILC and NST, albeit ILC had a higher initial CTC count (Figure 3). The downward trend continued but leveled out at three and six months of FU (Figure 3, Appendix A).

Higher CA 15-3 values (median, IQR) were observed in ILC than in NST cases both at BL (392, 1132 vs. 91, 230, *p* = 0.004) and after one month of systemic treatment (345, 650 vs. 61, 219, *p* = 0.007) (Table 2, Appendix A).

#### 3.2.2. Prognostic Information

The evidence for CTC ≥ 5 as a prognostic factor for PFS (HR 1.5, 95% CI 0.55–4.0, *p* = 0.44) and OS (HR 2.4, 95% CI 0.71–8.3, *p* = 0.16) in ILC cases was weak. In contrast, there was strong evidence of prognostic effects for the established cut off CTC ≥ 5 in the considerably larger subgroup of NST cases (PFS: HR 1.7, 95% CI 1.2–2.6, *p* = 0.007; OS: HR 2.1, 95% CI 1.3–3.3, *p* = 0.002; Figure 4 and Figure 5).

The prognostic impact of CTC count on PFS and OS in ILC cases was stronger with higher cut-offs (CTC ≥ 20: HR 3.0, 95% CI 1.3–6.8, *p* = 0.01, and HR 3.1, 95% CI 1.2–8.3, *p* = 0.02, respectively) (CTC ≥ 80: HR 3.6, 95% CI 1.5–8.8, *p* = 0.004, and HR 5.9, 95% CI 2.0–18, *p* = 0.002, respectively). The prognostic effect was essentially the same among NST cases for these higher cut-offs (Figure 4 and Figure 5). 

In a multivariable analysis adjusted for age, performance status according to the Eastern Cooperative Oncology Group (ECOG) criteria at BL, distant recurrence-free interval, number of metastatic sites (<3 vs. ≥3), and visceral vs. no visceral metastases, the prognostic CTC effect (cut-off ≥5) for PFS in ILC (HR 1.2, 95% CI 0.41–3.6, *p* = 0.72) and NST cases (HR 1.7, 95% CI 1.1–2.7, *p* = 0.02) remained essentially the same as in univariable analysis, as were the multivariable results for higher CTC cut-offs (≥20 and ≥80) (data not shown). 

In an exploratory landmark analysis on change in CTC count from ≥5 CTCs at baseline to <5 CTCs at 3 months of follow-up after initiation of systemic therapy, the observed change was associated with improved outcome in terms of PFS and OS, in comparison to patients with persistent CTCs ≥5 at 3 months, in ILC and NST (Appendix A). 

To further decipher the importance of dynamic changes in CTCs on outcome, we used Cox models with CTC status, updated during follow-up, as a time-dependent covariate in ILC and NST separately to predict outcome. For ILC, the progression incidence was 8.2 times higher during episodes with CTCs ≥5 compared with episodes with CTCs <5 (95% CI 2.5–26, *p* < 0.001). The corresponding HR for NST was 4.3 (95% CI 2.7–6.8, *p* < 0.001), but no evidence for differential effect of CTC-status on PFS for the two histopathological subtypes was seen (*p* for interaction = 0.69). 

The presence of one or more CTC clusters was a negative prognostic factor associated with impaired survival among ILC cases (PFS: HR 4.6, 95% CI 1.7–12, *p* = 0.003; OS: HR 4.9, 95% CI 1.7–14, *p* = 0.003), whereas the effect was weaker in NST cases (PFS: HR 1.2, 95% CI 0.69–2.0, *p* = 0.55; OS: HR 1.9, 95% CI 1.1–3.3, *p* = 0.02; Appendix A). The presence of CTC clusters in ILC cases was highly correlated with a CTC count ≥80, leading to multicollinearity problems in the Cox models that included both variables. Hence, the support for independent prognostic value of CTC clusters was weak in the present study.

The evidence for differences in PFS and OS was weak among both ILC and NST cases with a CA 15-3 cut-off of ≥30 U/mL. With higher cut-offs (≥100, ≥200, and ≥400 U/mL), stronger evidence for negative prognostic effects was observed for OS but not for PFS in both ILC and NST cases, with the most pronounced effect evident in NST cases (Appendix A). 

### 3.3. CTC and CA 15-3 as Monitoring Tools for Early Progression

For the patients with CTC data (ILC: *n* = 27 and NST: *n* = 99), diagnosis of early progression within 12 months was assessed. Early progression within 12 months was diagnosed by standard methods in 9/27 ILC cases, whereas 17 progressed after one year. The corresponding numbers for early progression in NST was 31/97 and 41 patients progressed after 12 months. During the first FU period (0–6 months) progressive disease was confirmed in only one ILC case by standard monitoring methods and this progression was not identified by concurrent CTC increase; but 18 out of 36 NST cases were identified by concurrent CTC evaluation. 

During the next FU period (>6–12 months), eight additional cases of ILC progression were diagnosed by standard monitoring methods and in four of them progression had been indicated by an increase in CTCs during 0–6 months of FU. The opposite was true for the NST cases, where none of the 13 cases of NST progression confirmed by standard monitoring methods during >6–12 months of FU could be identified by CTC evaluation during the first six months of FU.

In terms of CA 15-3 as a diagnostic tool for early progression (ILC: *n* = 27, and NST: *n* = 94), a total of 4 cases of ILC and 31 of NST progression were indicated by an increased level of CA 15-3 (17 concurrent with the standard monitoring methods and 14 by CA 15-3 only) during the first FU period (0–6 months). The only ILC case with progression during 0–6 months was not identified by increased CA 15-3 levels, and one out of eight cases of ILC progression during the following six months was indicated by a previous increase in CA 15-3 levels during the first six months of FU. One case of ILC-progression and 16 of NST-progressions, confirmed by standard monitoring methods during the first FU period, were not identified by concurrent CA 15-3.

## 4. Discussion

In this study, exploring CTCs and CTC clusters applying the CellSearch technique between metastatic ILC and NST cases in a prospective monitoring trial exclusively including patients scheduled for first-line systemic treatment, several distinguishing features were discovered. The CTC count at BL and presence of CTC clusters before the start of first-line treatment was remarkably higher among ILC cases. Despite this finding, we could for the first time show that the evidence for a prognostic value of the validated CTC cut-off (≥5) was weaker in this group. However, the prognostic impact was stronger in ILC with higher CTC cut-offs. The longitudinal design of the study enabled us to show a decline in CTCs and CTC clusters after one month of systemic treatment in both NST and ILC cases. Importantly, a decline in CTCs was translated into improved outcome in both ILC and NST compared with no change in CTC status. This finding supports the notion that a change in CTC status reflects the effect of systemic therapy even after longer follow-up time.

Previous publications on CTCs have, for the most part, investigated their distribution and prognostic effects in BC subtypes based on immunohistochemical (IHC) or gene expression assay classification, but differences related to histopathological type have, to a large extent, been overlooked. The present results are in line with those from a comparable retrospective CTC study [29], where CTC counts were significantly higher in metastatic ILC than in NST. In contrast to the present study, no survival data on ILC stratified by number of CTCs was presented. Since luminal A-like tumors (IHC classified by St. Gallen 2011 surrogate definitions) [37] with high CTC counts were overrepresented in the ILC subgroup, the authors hypothesized that the predominance of this subtype could be a confounder. Importantly, no multivariable analysis adjusting for the surrogate subtypes on the prediction of high CTC counts was reported. In the present study, we additionally adjusted the difference in CTC counts between the ILC and NST cases by analyses of the intrinsic PAM50 subtypes: luminal A vs. non-luminal A (luminal B + HER2-enriched + basal-like) MBCs [38] and found a consistent difference in the number of CTCs between ILC and NST. Another study [30] showed that the presence of one or more CTC was higher in the metastatic ILC as compared to the NST subgroup, again without providing any survival data. A higher presence of one or more CTC in cases of primary ILC (without distant metastases) than in NST was also reported [39]. These previous findings, along with our data on number of CTCs adjusted for the intrinsic subtypes, strengthen the probability that high CTC count is also dependent on the histopathological type rather than a merely luminal-related finding.

Classic ILCs are typically comprised of small, round, and loosely bound tumor cells with a characteristic single-file growth pattern [20]. A mutation or dysfunction in the *CDH1* gene is common (55%–100%), resulting in either a decreased expression or total loss of important cell–cell adhesion proteins included in the cadherin–catenin complex (E-cadherin and α-, β-, and γ-catenins) [22,40]. These characteristics may potentiate the ability of lobular (as opposed to ductal) tumor cells to detach from the primary tumor or metastases and enter the bloodstream, resulting in higher levels of CTCs. 

CTC detection by the CellSearch system is based on the use of epithelial cell adhesion molecule (EpCAM) for the capture and isolation of CTCs. In a study evaluating EpCAM expression in BCs, the levels were found to be negative to low in 74% of ILC and 49% of NST cases [41]. The low proportion of tumors with EpCAM expression in BC points out the limitation of the CellSearch system as a monitoring method in BC patients. Despite this limitation, the CellSearch technique is the only Food and Drug Administration (FDA) approved system for enumeration of CTCs. CellSearch enumeration provides clinically valid prognostic information, although it does not capture EpCAM negative CTCs. Interestingly, EpCAM positive cells have been implicated to harbor more prognostic information compared with EpCAM negative CTCs. The present results also support a potential risk for a lower CTC detection rate in ILCs than in NSTs. In this study, EpCAM expression was not analyzed, and the occurrence of a falsely low CTC distribution in ILCs could not be excluded. Regardless of the real conditions, the CTC count at BL was still higher in the ILC than in the NST cases. CTC capturing by labeling free methods such as microfluids and size-based selection have been developed but are yet not ready for clinical use. A study exploring the number of CTCs in metastatic ILC and NST with a label free method capturing cells with heterogenous phenotypes would be an interesting future area of investigation. 

Numerous CTC studies based on mixed histopathological BC types have shown significant prognostic effects when classifying MBCs with <5 CTCs as a better prognostic outcome (MBC_indolent_) and MBCs with ≥5 CTCs as worse prognostic outcome (MBC_aggressive_) according to the CellSearch technique [8,10]. This cut-off originates from the median CTC count in a seminal CTC study by Cristofanilli et al. [10]. Our analyses showed a negative prognostic CTC effect in both ILC and NST cases; however, the evidence for a prognostic value of the generally accepted CTC cut-off (≥5) was weaker in the substantially smaller ILC subgroup. A higher proportion of ILC than NST cases (79% vs. 46%, respectively) was classified as MBC_aggressive_, indicating a potentially worse prognosis for metastatic ILC. Despite this, the survival analyses showed similar prognosis for metastatic ILC and NST cases. A persistent CTC count ≥5 over time was, however, indicated to be more important for prediction of the outcome in ILC compared with NST. 

In agreement with previous studies exploring the prognostic value of higher CTC cut-offs in MBC [33,42], our results showed increasing prognostic effects associated with higher CTC cut-offs (≥20 and ≥80). In these analyses, the negative prognostic effect, as well as the evidence thereof, was stronger in the ILC subgroup, suggesting that a higher cut-off based on the median CTC count in ILC might be more suitable in ILC cases to better discriminate between the MBC_indolent_ and MBC_aggressive_ forms.

In this study, the presence of CTC clusters was more common in ILC than in NST cases. This finding is paradoxical, since previous studies suggested that CTC clusters have strong cell–cell contacts, held together through E-cadherin and catenin-dependent intercellular adhesion, where high levels of plakoglobin (γ-catenin) were identified as one of the most important factors for CTC cluster formation [12], whereas absence or dysfunction of these proteins is one of the cardinal features of ILCs [22]. Current CTC studies showed that clustering formation is based on a complex multifactorial process and also that the CTC cluster microenvironment is essential, but the exact mechanisms are still largely unknown [12,43,44,45]. Our findings raise new questions: Are there different underlying mechanisms responsible for CTC clustering, and are there important histopathological differences in the CTC cluster microenvironment in ILCs and NSTs? Further studies exploring CTC clusters in metastatic ILC and NST cases are thus encouraged.

The distributional differences in CTCs and CTC clusters at BL between ILC and NST cases were, to a large extent, equalized when evaluated after one month of FU. This finding suggested that systemic treatment provides a measurable decline in the ILC subgroup. The underlying cause thereof still needs to be further investigated; however, this could potentially be a clinically useful early indicator of treatment effect, and we demonstrated that an early decline in CTC count from baseline to three months is related to a more favorable outcome. The systemic first-line MBC treatments in this study were well balanced (endocrine therapy vs. chemotherapy) (Table 1), suggesting that the differences in CTC and CTC cluster decline were not related to differences in endocrine or chemotherapeutic sensitivity between the subgroups. 

Despite a generally higher CTC count and the presence of CTC clusters, ILC cases displayed a lower metastatic burden with fewer metastatic sites and a higher degree of solitary bone metastases. Hypothetically, differences in CTC biology could play a role and the micro metastatic environment in skeletal metastases might favor CTC shedding and cluster formation. Based on our observations, it appears likely that lobular as opposed to ductal CTCs and CTC clusters more easily enter the circulation, although their ability to survive and subsequently form metastases is inferior. The mechanisms behind this are hitherto unclear. Moreover, the collinearity of clusters with number of CTCs in this study made any independent prognostic assessment of clusters difficult. However, the metastatic potential of these CTCs remains to be established. Exploratory studies searching for biomarkers capable of discriminating the metastatic potential in single CTCs and CTC clusters are warranted.

In this study, we focused on early progression in metastatic disease to evaluate CTCs and CA 15-3 as monitoring tools for a timely change of therapy. Monitoring is especially challenging in patients with unmeasurable disease exemplified by skeletal only metastases. ILCs are characterized by a high frequency of bone metastases and metastatic dissemination to atypical sites which are considered more difficult to evaluate with standard monitoring. CTC enumeration by CellSearch has been indicated as a promising candidate for treatment monitoring, although the clinical utility has been difficult to prove given the fact that not all patients are CTC positive. In the present study, an early detection of progressive disease prior to standard monitoring methods by CTCs provided no conclusive results due to the small sample size and the study design where CTC and CA 15-3 sampling did not continue beyond six months of FU. The same was true using CA 15-3 levels as a monitoring tool.

The improvement of liquid biopsy techniques including cell free DNA could hopefully be developed into a real-time monitoring tool in the future, capturing signs of early progression in all patients and enabling a more rapid change of therapy.

The strengths of this study include its prospective monitoring design, with a predefined protocol, and the composition of the cohort including only previously untreated patients with MBC scheduled for first-line systemic treatment. Few patients were lost to FU, and the evaluation process, with serial CTC sampling, was executed based on validated state-of-the-art techniques. Limitations include the relatively small study sample size, indicating that the study was not powered to detect clinically relevant prognostic differences among ILC cases. No phenotypic characterization of CTCs and CTC clusters was performed. Since liquid biopsy sampling ceased at six months of FU and a majority of the study patients developed progressive disease after six or more months of FU, the possibility to explore CTCs and CA 15-3 levels as monitoring factors was limited.

## 5. Conclusions

This study showed that there are different distributional and prognostic CTC features in metastatic ILC and NST cases applying the CellSearch technique. The number of CTCs and CTC clusters was higher in ILC than in NST cases before the start of first-line systemic treatment, and these results implied that a higher CTC cut-off could be considered in metastatic ILC cases. Finally, eradication of CTCs from BL to three months predicted favorable long-term outcome and indicated treatment response in both ILC and NST cases.

## Figures and Tables

**Figure 1 cells-09-01718-f001:**
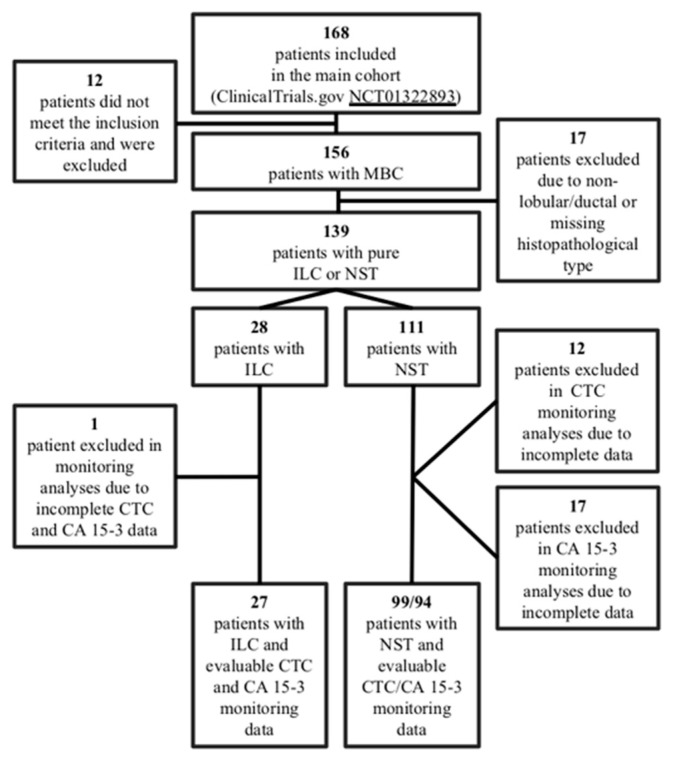
Flowchart of the study cohort.

**Figure 2 cells-09-01718-f002:**
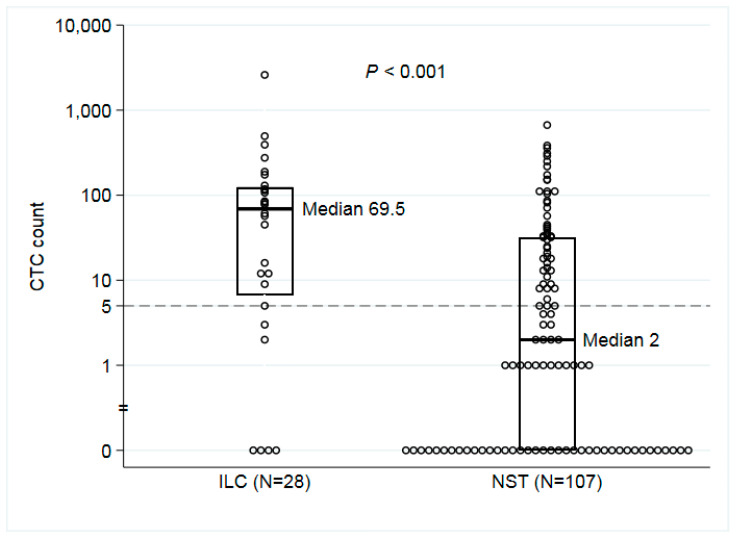
Circulating tumor cell (CTC) count at baseline by histopathological type. Abbreviations: ILC, invasive lobular carcinoma; NST, invasive ductal carcinoma of no special type; CTC count, number of CTCs per 7.5 mL blood.

**Figure 3 cells-09-01718-f003:**
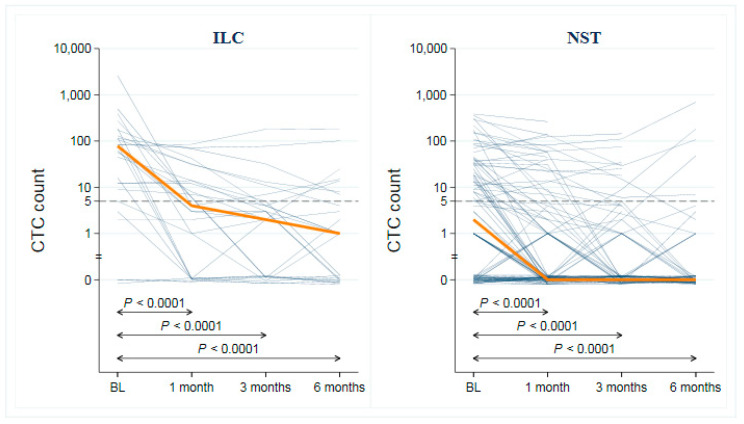
Distribution of CTC counts at baseline and with serial sampling at different time points. Spaghetti plots showing number of CTCs (per 7.5 mL blood) per patient from baseline (BL) to 6 months of follow-up for invasive lobular carcinoma (ILC), left panel (**A**), and invasive ductal carcinoma of no special type (NST), right panel (**B**). The *p*-values correspond to pairwise tests of the null hypothesis of no change in CTC count (Wilcoxon matched-pairs signed-rank test). Note that the scale on the y-axis is logarithmic and that the y-axis has been broken to enable presentation of zeros. A small amount of random noise was added to all zeros to separate the lines. The red line connects the medians at the four time points.

**Figure 4 cells-09-01718-f004:**
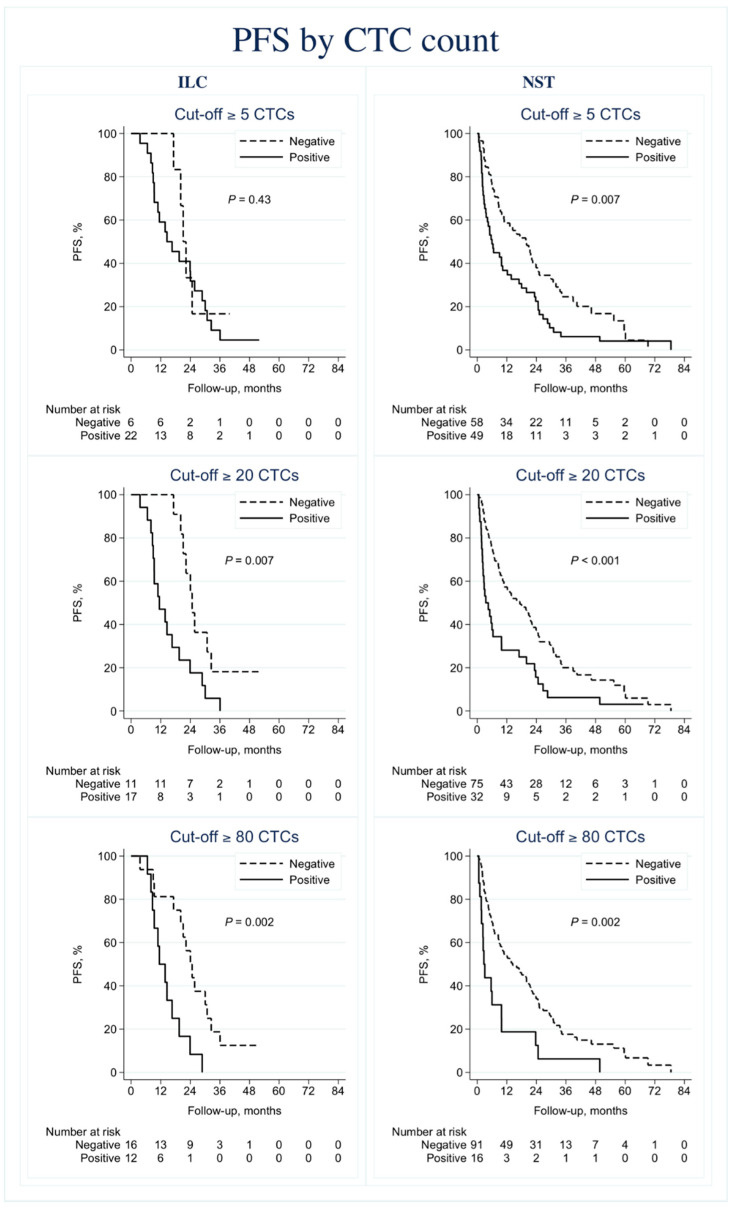
Progression-free survival (PFS) by circulating tumor cell (CTC) count. Kaplan–Meier plots displaying PFS for the invasive lobular carcinoma (ILC) and invasive ductal carcinoma of no special type (NST) subgroups. Cut-off ≥5 CTCs (**A**–**B**). Cut-off ≥20 CTCs (**C**–**D**). Cut-off ≥80 CTCs (**E**–**F**).

**Figure 5 cells-09-01718-f005:**
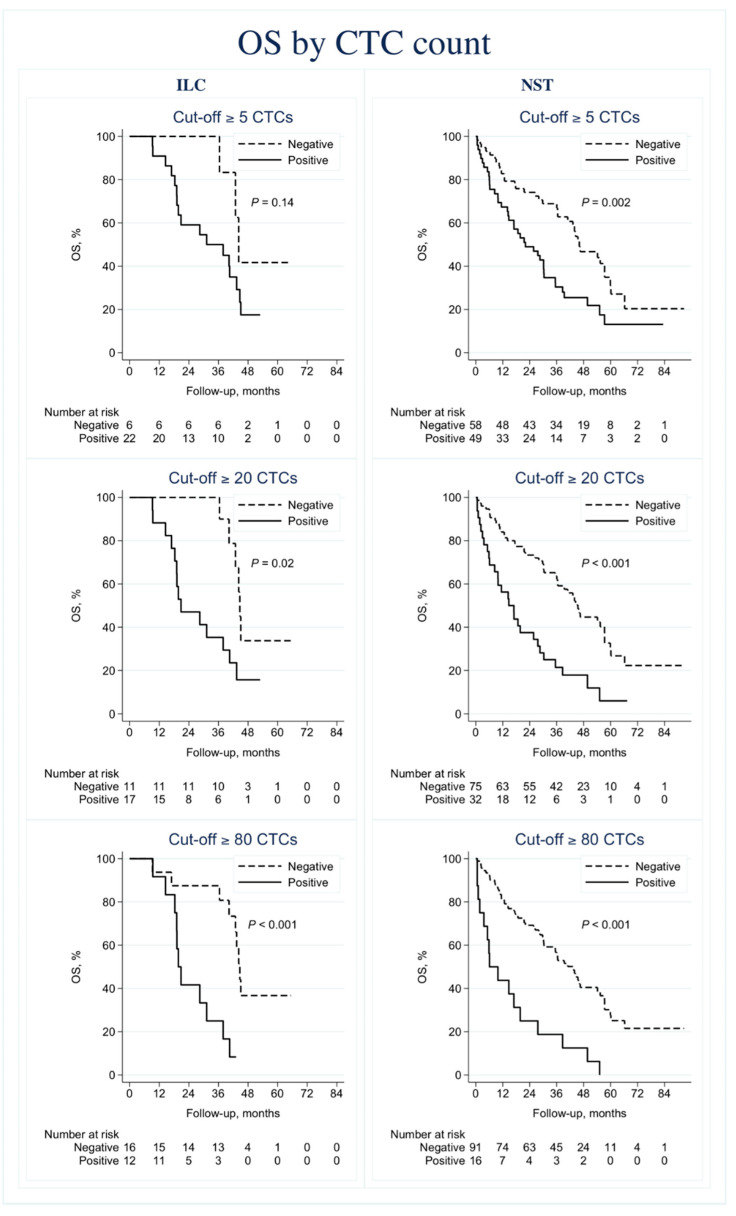
Overall survival (OS) by circulating tumor cell count. Kaplan–Meier plots displaying OS for the invasive lobular carcinoma (ILC) and invasive ductal carcinoma of no special type (NST) subgroups. Cut-off ≥5 CTCs (**A**,**B**). Cut-off ≥20 CTCs (**C**,**D**). Cut-off ≥80 CTCs (**E**,**F**).

**Table 1 cells-09-01718-t001:** Baseline patient and tumor characteristics by histopathological type.

Variables	ILC (*n* = 28) (%)	NST (*n* = 111) (%)	ILC vs. NST *p*-Value
**Age at MBC Diagnosis, (years) Median**	67 (range 47–82)	65 (range 40–90)	0.36 ^f^
**Baseline Performance Status (ECOG)**			
0	19 (68)	56 (53)	0.17 ^g^
1	6 (21)	30 (29)	
2	3 (11)	19 (18)	
Unknown	0	6	
**NHG** ^a^			
I	1 (5)	9 (10)	**0.006** ^g^
II	21 (95)	40 (44)	
III	0 (0)	42 (46)	
Unknown	6	20	
**Tumor Size** ^A^			
T1	7 (33)	42 (40)	0.79 ^g^
T2	13 (62)	36 (34)	
T3	4 (19)	12 (11)	
T4	1 (5)	16 (15)	
Unknown	3	5	
**Axillary Node Status** ^A^			
Negative	2 (8)	35 (36)	**0.008** ^g^
Positive	22 (92)	62 (64)	
Unknown	4	14	
N0	2 (8)	35 (36)	**0.002** ^h^
N1 (1–3)	10 (42)	32 (33)	
N2 (4–9)	4 (17)	22 (23)	
N3 (>9)	8 (33)	8 (8)	
**IHC Phenotype** ^b^			
HR+ HER2–	23 (82)	71 (66)	0.27 ^g^
HER2+	2 (7)	15 (14)	
HR– HER2–	3 (11)	21 (20)	
Unknown	0	4	
**PAM50 Subtype** ^c^			
Luminal A	15 (71)	28 (31)	**0.006** ^h^
Luminal B	4 (19)	36 (40)	
HER2-enriched	2 (10)	13 (14)	
Basal-like	0 (0)	13 (14)	
Unknown	7	21	
**Distant Recurrence Free Interval (Years)**			
Median (range)	5.3 (0–36)	4.7 (0–24)	0.41 ^f^
0	6 (21)	21 (19)	0.81 ^g^
>0–3	3 (11)	22 (20)	
>3	19 (68)	68 (61)	
**MBC at Primary Diagnosis**			
Yes	6 (21)	21 (19)	0.76 ^g^
No	22 (79)	90 (81)	
**No. of Metastatic Sites**			
<3	24 (86)	76 (68)	0.07 ^g^
≥3	4 (14)	35 (32)	
**Site of Metastasis**			
Visceral ^d^	8 (29)	72 (65)	**0.001** ^g^
Non-visceral	20 (71)	39 (35)	
Bone only	11 (39)	24 (22)	0.05 ^g^
Not bone only	17 (61)	87 (78)	
**1st Line Treatment for MBC** ^e^			
Endocrine	13 (46)	43 (39)	0.90 ^h^
Chemotherapy only	13 (46)	57 (52)	
HER2-targeted + Chemotherapy	2 (7)	9 (8)	
Unknown	0	2	
**Progression-Free**			
Yes	2 (7)	10 (9)	N/A ^i^
No	26 (93)	101 (91)	
**Alive**			
Yes	8 (29)	34 (31)	N/A ^i^
No	20 (71)	77 (69)	

^a^ Based on data from the primary tumor and axillary lymph nodes at the time of primary breast cancer diagnosis. ^b^ Immunohistochemistry (IHC) phenotype was primarily derived from IHC staining of the metastasis (*n* = 105). If no information was available from the metastasis, the phenotype was derived by staining of the primary tumor (*n* = 30). ^c^ PAM50 subtypes were derived from the primary tumor. ^d^ Visceral metastasis defined as lung, liver, brain, peritoneal, and pleural involvement. ^e^ Treatment decision was based on IHC phenotype. Eleven patients died or treatment was ended before the first structured clinical follow-up at 3 months post treatment initiation. No data available for these patients. ^f^
*p*-value from Mann–Whitney U test. ^g^
*p*-value from Pearson´s chi-square test (linear by linear association test if more than two ordered categories). ^h^
*p*-value from Fisher´s exact test. ^i^ Not applicable. Significant *p*-values are presented in bold font. Abbreviations: ILC, invasive lobular carcinoma; NST, invasive ductal carcinoma of no special type; CTC, circulating tumor cell; MBC, metastatic breast cancer; ECOG, Eastern Cooperative Oncology Group; NHG, Nottingham histological grade; PT, primary tumor; HR, hormone receptor; HER2 = human epidermal growth factor receptor 2.

**Table 2 cells-09-01718-t002:** Baseline CTC and CA 15-3 distributions.

Baseline CTC Status ^a^ (No. Per 7.5 mL Blood)	ILC (*n* = 28) (%)	NST (*n* = 111) (%)	ILC vs. NST *p*-Value
CTC Median (Range)	70 (0–2598)	2 (0–668)	**<0.001** ^c^
CTC < 5	6 (21)	58 (54)	
CTC ≥ 5	22 (79)	49 (46)	**0.003** ^d^
CTC < 20	11 (39)	75 (70)	
CTC ≥ 20	17 (61)	32 (30)	**0.004** ^d^
CTC < 80	11 (39)	91 (85)	
CTC ≥ 80	17 (61)	16 (15)	**0.003** ^d^
CTC clusters absent	18 (64)	88 (82)	
CTC clusters ≥ 1	10 (36)	19 (18)	0.07 ^d^
Unknown	0	4	
**Baseline CA 15-3 value**^b^ (U/mL)			
CA15-3 Median (Range)	392 (17–2999)	91 (6–2999)	**0.004** ^c^
CA 15-3 < 30	4 (15)	25 (24)	
CA 15-3 ≥ 30	23 (85)	79 (76)	0.30 ^d^
CA 15-3 < 100	8 (30)	56 (54)	
CA 15-3 ≥ 100	19 (70)	48 (46)	**0.03** ^d^
CA 15-3 < 200	11 (41)	73 (70)	
CA 15-3 ≥ 200	16 (59)	31 (30)	**0.004** ^d^
CA 15-3 < 400	14 (52)	84 (81)	
CA 15-3 ≥ 400	13 (48)	20 (19)	**0.002** ^d^
Unknown	1	7	

^a^ Four patients had no CTC or CTC cluster data available due to missing BL sample. ^b^ Eight patients had no CA 15-3 data available due to missing BL sample. ^c^
*p*-value from Mann–Whitney U test. ^d^
*p*-value from Pearson´s chi-square test. Significant *p*-values are presented in bold font.

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
