# Peer review of "The Distribution of Circulating Tumor Cells Is Different in Metastatic Lobular Compared to Ductal Carcinoma of the Breast—Long-Term Prognostic Significance"

_cells, 2020, doi:10.3390/cells9071718_

Round 1

Reviewer 1 Report

The authors have addressed the comments I made earlier.  The manuscript is acceptable for publication.

Reviewer 2 Report

The authors have addressed all my concerns

This manuscript is a resubmission of an earlier submission. The following is a list of the peer review reports and author responses from that submission.

Round 1

Reviewer 1 Report

Review of Narbe et al.

The manuscript by Narbe et al. focuses on CTCs in breast cancer and compares the numbers of CTCs during follow-up collection for invasive lobular carincoma (ILC) and invasive ductal carcinoma of no special type (NST). The authors compare the CTC counts they find suing Cell Search with the commonly used breast cancer biomarker CA 15-3.

This is an interesting approach and the authors should be complimented on tackling the different breast cancer sub-types and their challenges in predicting PFS and OS for the individual patient.

In the present form, however, the study has serious issues.

  • As the authors indicate, the numbers of CTCs collected using Cell Search is dependent on EpCAM. A previous study (also cited by the authors of this manuscript) identified low or no EpCAM expression in 74% ILC and 49% of NST (Soysla et al., 2013). This fact puts serious doubts on the reliability of the CTC capture and suggests ‘selective’ capture of certain CTCs only. How can we build a monitoring scheme when what we need to monitor is not captured? In other words, if CTC counts are based on a maximal capture of 26-51% of the CTCs population present, how reliable are these counts? Should not a non-EpCAM-based technology be used?
  • The statement by the authors “ A study exploring the number of EpCAM-positive vs. EpCAM-negative CTCs in metastatic ILC and NST would be an interesting future area of investigation” (page 13 of 18) is not adequate, as prior knowledge already indicates that there is a wide heterogeneity, and this impacts on Cell Search’s identification of the cells.
  • The conclusions that this study “… showed that there are different distributional, prognostic, and monitoring CTC features in metastatic ILC and NST cases. “ is an over-interpretation due to the flaw in the CTC capture procedure.
  • No phenotypic or molecular characterization of the CTCs was performed.

Minor comments:

Section 3.3. needs work: it lacks clarity in the description of the results. One Table combining ILC and NST is preferable.

Reviewer 2 Report

The authors isolated CTC from peripheral blood of 139 patients with metastatic breast cancer and compared progression of ILC and NST in relation to common standard methods.

Several points should be addressed:

  1. The authors use very old references in the introduction for annual data.
  2. Authors reference the inclusion and exclusion criteria for participants of the study from another publication. Criteria should be clearly explained in the manuscript.
  3. Data of FU for NST is very inconsistent and shows no CTC after treatment on average.
  4. If median is zero, evaluation is not meaningful and numbers should be displayed in detail not only as cut off <5. Additionally, the number of CTCs at BL varies unreasonable for NST.
  5. The reviewer does not agree with the conclusion that decline of CTCs was more pronounced in ILC. For NST, the authors show zero CTCs in median for FU what is not comparable for a decline.
  6. How do the authors explain that others indicate the cut off <5 CTC as prognostic and their results do not?
  7. Several interpretations based on one single patient which is not sufficient for prognosis (e.g. early progression).
  8. The authors explain that figure 2 shows adjustment of PAM50 subtype. This is not obvious.
  9. Figure 6 and 7 display missing data. Why was evaluation of CTCs not performed for complete FU of more than 6 months?
  10. No information is available for specific patient treatment or uniformity.
  11. Metastatic sites differ in ILC and NST. Detailed discussion of this fact is missing.
  12. What about the number of CTCs after 6 months? Median FU time was 49 months.
  13. The authors declare that 30% of the patients survived the FU but Kaplan-Meier plots do not display this number of patients for ILC and NST.
  14. The authors do not intensively investigate CTC clusters but speculate about the meaning. This should be more intensified.

Reviewer 3 Report

The authors have presented a nice study looking at differences in CTC count between ILC and NST patients. Overall, i have no real concerns, only one being what markers were used to detect CTCs? Also, Figure 1 could be made a bit clearer.